# Gender Differences in Chronic Hormonal and Immunological Responses to CrossFit^®^

**DOI:** 10.3390/ijerph16142577

**Published:** 2019-07-19

**Authors:** Rodrigo Poderoso, Maria Cirilo-Sousa, Adenilson Júnior, Jefferson Novaes, Jeferson Vianna, Marcelo Dias, Luis Leitão, Victor Reis, Nacipe Neto, José Vilaça-Alves

**Affiliations:** 1Physical Education Department, University of Unopar, Nilópolis 36045–050, Brazil; 2Physical Education Department, Federal University of Paraíba, João Pessoa 58051–900, Brazil; 3Posgraduate Program of Educação Física, University of Cariri Regional, Crato 63105–010, Brazil; 4Federal Technology Institute of Paraíba, João Pessoa 58015–020, Brazil; 5Gymnastics Department, Federal University of Rio de Janeiro, Rio de Janeiro 21941–901, Brazil; 6Faculty of Physical Education Sports, Federal University of Juiz de Fora, São Pedro 36036-900, Brazil; 7Granbery Methodist College, Juiz de Fora 36015–440, Brazil; 8Superior School of Education of Polytechnic Institute of Setubal, Setúbal 2910–761, Portugal; 9Research Centre in Sports Sciences, Health Sciences and Human Development, CIDESD, Vila Real 5000–103, Portugal; 10Faculty of Medical and Health Sciences of Juiz de Fora, Suprema, Juiz de Fora 36033–003, Brazil; 11Division of Endocrinology, IPEMED Medical School, Rio de Janeiro 22031–060, Brazil; 12Sports Department, University of Trás-os-Montes and Alto-Douro, Vila Real 5001–801, Portugal

**Keywords:** CrossFit, hormonal responses, immunological responses, RPE

## Abstract

This study was designed to analyze the chronical responses of the hormonal and immune systems after a CrossFit^®^ training period of six months as well as to compare these results between genders. Twenty-nine CrossFit^®^ practitioners (35.3 ± 10.4 years, 175.0 ± 9.2 cm, 79.5 ± 16.4 kg) with a minimum CrossFit^®^ experience of six months were recruited, and hormonal and immune responses were verified every two months during training. The training was conducted in five consecutive days during the week, followed by two resting days. Testosterone (T) values were significantly higher at the last measurement time (T6 = 346.0 ± 299.7 pg·mL^−1^) than at all the other times (*p* < 0.002) and were higher in men than in women (*p* < 0.001). Cortisol (C) levels were lower at all times compared to the initial level before training, and differences were observed between men and women, with men having a lower value (T0: *p* = 0.028; T2: *p* = 0.013; T4: *p* = 0.002; and T6: *p* = 0.002). The TC ratio in women was lower at all times (*p* < 0.0001) than in men. Significant effects on CD8 levels at different times (F_(3.81)_ = 7.287; *p* = 0.002; η*_p_*^2^ = 0.213) and between genders (F_(1.27)_ = 4.282; *p* = 0.048; η*_p_*^2^ = 0.137), and no differences in CD4 levels were observed. CrossFit^®^ training changed the serum and basal levels of testosterone and cortisol in men (with an increase in testosterone and a decrease in cortisol).

## 1. Introduction

CrossFit^®^ is seen as an alternative to high-intensity functional training (HIFT). Despite being relatively new, this training and competition program is becoming increasingly popular across the world. CrossFit^®^ programming is based on constantly varied functional movements performed at relatively high intensity and includes metabolic conditioning, gymnastics, weightlifting [1]. Despite different CrossFit^®^ training sessions varying in their exercises and motion patterns, they are all characterized by the use of high-intensity exercises with little to no resting periods in order to complete a task as fast as possible (AFAP) or achieve the greatest number of repetitions in a certain period of time (as many rounds/repetitions as possible, AMRAP). The training loads used are not individualized, which can be excessive for some individuals and lead to an increased risk of injury [2].

The high intensity of CrossFit^®^ workouts leads to perceptive, hemodynamic [3], metabolic [4], and hormonal [5] responses in the organism of the participants. According to Fernandez-Fernandez et al. [3], the intensity of these exercises is characterized by a heart rate (HR) close to 95% of the maximum HR and a rate of perceived exertion (RPE) higher than 8 in a 0–10 scale. Due to this fact, studies have observed increased lactate concentrations [4,6], oxidative stress [7], inflammation [8,9], and levels of immune responses biomarkers and serum hormones [5,10].

Regarding the hormonal responses, Mangine et al. [10] have evaluated the acute testosterone and cortisol responses across five weeks of CrossFit^®^ open trainings. The authors observed significant changes in the serum levels of testosterone and cortisol, which were affected in different ways at each workout depending on the overload and workout duration. These results comply with previous studies that have been conducted with marathon runners [11] and resistance training practitioners [12].

CrossFit^®^ practitioners may also present acute responses of oxidative stress, which influences their immune system [7]. Jin et al. [13] have observed that intense exercise has an immunosuppressive effect if resting periods and an adequate nutritional intake are not appropriate. The authors found that people who practiced resistance training with loads above 85% of their one repetition maximum (1RM) or cyclical exercises of cardiorespiratory endurance with an intensity around 85% of their maximum rate of oxygen consumption (VO_2 max_) presented a decrease of the T-helper (CD4) and cytotoxic (CD8) lymphocytes post-workout, compared with their resting values.

Few studies have compared these responses to CrossFit^®^ training in men and women. Murawska-Cialowicz et al. [14] measured at rest changes in brain-derived neurotrophic factor and in irisin levels and improvements in aerobic capacity and body composition of young physically active men and women. However, similar results were observed for testosterone (T) and cortisol (C) levels and T/C ratio in men and women and when evaluating the responses with respect to gender and time of exercise.

After reviewing the current scientific literature related to HIFT, we concluded that no studies have yet analyzed the chronical hormonal (testosterone and cortisol) response in CrossFit^®^ practitioners. Furthermore, no sufficient data were found in relation to T lymphocytes alterations during exercise. Therefore, it seemed pertinent to study the chronical hormonal and immune responses in CrossFit^®^ practitioners and possible gender differences. Thus, the aim of this study was to analyze the chronical responses of the hormonal and immune systems after a CrossFit^®^ training period of six months and to compare these results between female and male practitioners. We hypothesized that (a) there would be changes in the chronical responses of hormonal and immune systems after a six-months CrossFit^®^ training intervention, (b) only men would present changes in the chronical hormonal responses, and (c) no differences would be observed between genders in chronical immunological responses.

## 2. Materials and Methods

### 2.1. Sample

Twenty-nine CrossFit^®^ practitioners (35.3 ± 10.4 years, 175.0 ± 9.2 cm, 79.5 ± 16.4 kg) with a minimum experience of six months were recruited for this study. Using the Gpower software (Heinrich-Heine-University of Düsseldorf, Düsseldorf, Germany) to apply ANOVA repeated measures (F test), we determined the ideal sample size as being 17 men and 12 women [15].

The sample respected the following inclusion criteria: not smoking, not drinking alcoholic beverages or moderate drinking, having no history of muscle or joint injury, not taking any medicines, anabolic steroids, or similar substances, such as any kind of nutritional supplements, and have a minimum program attendance of 85%. All volunteers signed an informed consent form in accordance with the Helsinki Declaration (1964) and the Nuremberg Code (1947) concerning research involving human subjects.

### 2.2. CrossFit^®^ Training Intervention

This study was conducted during a period of six months, with hormonal and immune responses being verified every two months of training. Blood samples were taken at the beginning (T0) of the training period and every two months afterwards (T2, T4, and T6). The training protocol which was used for this study followed the CrossFit^®^ model [1]. The training was conducted in five consecutive days during the week, followed by two resting days. The training sessions that were assigned were constantly varied and maintained a combination of metabolic conditioning, weightlifting, and gymnastic modalities (see Table 1)

All training sessions were held in a training center affiliated with CrossFit Inc. and supervised by a CrossFit^®^ Level 1 certified trainer. Each training session lasted approximately 60 min, consisting of a joint mobility period, a warm-up, a technical part, a workout of the day (WOD), and a cooling down period. Previous research has shown that in order to have significant effects on the various body structures and functions, a minimum of 16 CrossFit^®^ sessions during a period of three to five weeks is needed [16,17]. This study used a chronic period of 120 sessions. To the best of our knowledge, no other study has applied such training volume. Subjective effort perception was used in order to control the intensity of each training session. The participants were asked to refrain from any physical exercise outside the study but were otherwise allowed to maintain their usual daily diets and life style.

### 2.3. Rating of Perceived Exertion (RPE)

It was established that the subjects should maintain an RPE between 8 and 10 [3] in the OMNI-Resistance Exercise Scale (OMNI-RES) 0–10 scale [18]. RPE was controlled after each WOD and immediately after the end of the training sessions. Before the beginning of the study, the participants were given a copy of the instructions regarding the OMNI-RES scale [19]. The values for high and low perceived exertion were established through recall, according to the process described by Robertson [20].

### 2.4. Blood Samples and Analysis

Blood samples were collected twice at each measurement time (T0, T2, T4, T6), once in the morning, to control for any variations due to the hormonal circadian rhythm, and 12 h after training. For each sample, 5 mL of blood was drawn from the antecubital vein by a qualified professional and sent to a specialized medical laboratory for analysis. The samples were transferred into plastic test tubes and carried in an isolated box, and the serum was isolated and stored at −4 °C until the analysis. In order to assess changes in the plasmatic hormonal levels, chemiluminescent assay kits were used for testosterone determination [20], and radioimmunoassay kits were used for cortisol determination [21]. CD4 and CD8 lymphocytes were measured by flow cytometry [13].

### 2.5. Statistical Analysis

In order to describe the central tendency and variability of the different parameters, an initial exploratory analysis of the data was conducted. A graphical representation of all variables was used in order to detect any possible outliers or incorrect data entry. In order to calculate inferential statistics for the data, the normality of the distribution was assessed through the Shapiro–Wilk test, and the homogeneity and sphericity were tested through the Levene and Mauchly tests, respectively.

For testosterone and cortisol, the data did not meet the inferential parametric assumptions. Thus, a non-parametric method, the Wilcoxon’s test, was used for the analysis between measurement times, and the Man–Whitney test for the comparison between genders at each measurement time. After verifying the assumptions for the use of parametric tests for the CD4 and CD8 variables, univariate analysis of variance (ANOVA) was performed for the T0 time to observe the existence of statistically significant differences between participants. A two-way repeated measure (gender vs moments) ANOVA was performed, followed by a post-hoc Bonferroni test. The effect size measures were presented through partial eta squared (η*_p_*² value), with cut-off points of 0.10, 0.25, 0.40 representing small, medium, and high effect, respectively [22]. The significance level was set at *p* < 0.05. The data analysis was performed with SPSS (Statistical Package for the Social Sciences, SPSS Science, Chicago, IL, USA) statistics software.

## 3. Results

### 3.1. Subjects

Twenty-nine people participated in the study, 17 men and 12 women, with an attendance rate of 80%. Both men and women had similar ages and CrossFit^®^ experience, differing in height, total body weight, and fat percentage (see Table 2).

### 3.2. Testosterone

In total, significant differences were observed between measurement times, with testosterone values at T6 being significantly higher than at T0 (32.22%), T2 (19.52%), and T4 (15.84%). At all times, testosterone values were significantly higher in men (*p* < 0.001). In men, testosterone levels at T6 (*p* < 0.0001) were higher in comparison with all other times (34.07%; 25.09%; and 19.21%). No significant effect was observed in women (see Table 3).

### 3.3. Cortisol

There was a significant effect of CrossFit^®^ training on cortisol concentrations (*p* = 0.002) at different times. Cortisol levels at T6 were significantly lower than at T0 and T2 (18.89%; and 19.34%, respectively). Cortisol levels at T4 were significantly lower than at T0 (13.33%). Significant differences were observed between men and women, with cortisol levels in men being lower at T0 (30.97%), T2 (36.97%), T4 (32.14%), and T6 (31.15%). In men, there were significant differences between times (*p* = 0.025), with T4 values being significantly lower than T0 values (14.74%), and T6 values being lower than T0 (19.23%) and T2 (16%) values. There was not a significant difference in cortisol levels at different times in women (Table 3).

### 3.4. Testosterone/Cortisol Ratio (TC)

CrossFit^®^ training had a significant effect on TC both when comparing different measurement times moments (*p* < 0.001) and when comparing men and women (*p* < 0.001). The TC ratio in women was significantly lower at all times (*p* < 0.0001) than in men. In men, the TC Ratio at T0, T2, and T4 was also significantly lower than at T6 (36.13%, *p* = 0.00075; 37.57%, *p* = 0.00036; 25.93%, *p* = 0.00037, respectively) (Table 3).

### 3.5. CD4 and CD8 T Lymphocytes

Table 4 shows that there was no significant effect on CD4 levels. However, there was a significant effect on CD8 levels in time (F_(3.81)_ = 7.287; *p* = 0.002; η*_p_*^2^ = 0.213) and with respect to gender (F_(1.27)_ = 4.282; *p* = 0.048; η*_p_*^2^ = 0.137). CD8 levels were significantly higher at T6 in comparison with T2 and T4 (*p* < 0.0001; CI 95% = (43.9, 144.9); and *p* = 0.028; CI 95% = (3.7, 91.5), respectively). Women presented lower values of CD8 at T2 than men (*p* = 0.017; CI 95% = (−335.9, −36.6)).

In men, a significant effect on CD8 levels was observed at different times (F_(3.48)_ = 2.824; *p* = 0.049; η*_p_*^2^ = 0.150), with the CD8 values at T6 being significantly greater than at T2 and T4 (*p* = 0.024; CI 95% = (7.2, 130.5); and *p* = 0.023; CI 95% = (6.1, 104.1), respectively). In women, an effect was observed at different times (F_(3.33)_ = 4.925; *p* = 0.026; η*_p_*^2^ = 0.309), with T6 values being significantly greater than T2 values (*p* = 0.015; CI 95% = (21.1, 218.8)). No significant differences were observed in time for CD4, both in men and in women.

## 4. Discussion

This study started from the assumption that there are chronic responses of the hormonal and immune systems to six months of CrossFit^®^ training. In this study, there was an increase in testosterone levels after six months of training (T6) in comparison with the previous times (T0, T2, and T4). This increase can be due to the small rest intervals that occur during the training. Furthermore, Mangine et al. [10] have stated that an increase in exercise load combined with an increase in the volume of exercises and body weight can produce an augmented response in testosterone. In a study by Velasco-Orjuela et al. [23], which analyzed the impact of high-intensity interval training (HIIT), resistance training, and a combination of both, the HIIT group experienced a decrease in cortisol and an increase in testosterone.

In the present study, it was found that there was an increase in testosterone as well as a decrease in cortisol after six months of CrossFit^®^ training. This is consistent with earlier studies that reported a significative elevation in testosterone levels with high-intensity aerobic exercises (HIT) [24] and HIIT [25]. In this regard, Kraemer and Ratamess [26] stated that high-volume and high-intensity protocols with short rest intervals tend to lead to greater hormonal elevations. Recently, Mangine et al. [10] reported an elevation in testosterone in CrossFit^®^ training participants that may be due to transitory elevations in muscular force-producing capabilities. It may be that the periodization in CrossFit^®^ has an influence depending on the combination of the different training variables (exercise type and modality, volume and load) over time. Training overload, as well as motor unit recruitment, throughout the different exercises related to Olympic weightlifting and powerlifting may contribute to these elevations.

Cortisol concentration was inversely proportional to testosterone concentration. França et al. [11] found that the levels of both testosterone and cortisol change according to the intensity and duration of exercise. A decrease in cortisol concentrations, especially in men, can be a chronical adaptation to exercise. Mangine et al. [10] stated that recreationally active individuals experience adaptive organic developments to protect the muscles and other tissues sensible to glucocorticoids and avoid damaging effects.

Prolonged cortisol elevations negatively affect skeletal muscles and, consequently, exercise performance. Previously, a study [12] analyzed the TC ratio following exercise and observed a decrease of around 30% in this ratio in elite weightlifters after a year of a training protocol. A recent study found similar results, with a >30% decrease of the TC ratio 60 min after training in the first week and 30 min after training in the fifth week [10]. As the previous study used different training exercises every week, it may seem that the different responses are regulated by different training overloads. However, in this study, no differences were observed in the TC ratio during six months of training. This may be due to the fact that CrossFit^®^ follows a continuously varied regimen regarding the training load and volume. These are only preliminary findings, and further studies should be conducted, considering additional anabolic status and overtraining markers.

Significant effects were observed in CD8 T lymphocytes concentrations, whereas for CD4 no significant effects or interactions were observed. Krinski et al. [27] found that high-intensity and long-duration exercises induce immediate lymphocytosis, occurring in a transitory fashion and returning to the resting state after a short post-exercise period (3 to 72 h), which will depend on the applied intensity.

Various studies have shown an increase in CD8 during cyclic exercises including cycling [28] and running [29]. During CrossFit^®^, a study [4] has assessed two different training sessions and found that only one session was able to generate a stimulus that promoted important metabolic changes, with the decrease in the levels of anti-inflammatory cytokines, such as IL6 and IL10. Lastly, it seems that oxidative stress has to do more with the intensity than with the modality of exercise. Kliszczewicz et al. [7] observed similar results in oxidative stress biomarkers after one CrossFit^®^ training session in comparison with high-intensity training on a treadmill.

Jin et al. [13] demonstrated that both resistance training and an intense race can lead to a temporary immunosuppression. Both types of training led to an increase in the CD8 cells and not in CD4 cells, which suggests a state of temporary immunosuppression, with the lymphocyte serum concentration returning to the resting values after 30 min of resting. Shiu et al. [30] showed that, in healthy men from the Canadian Armed Forces, there was a decrease in CD4 and CD8 T cells concentration after a session of HIIT, returning to the resting values after 60 min, whereas after two weeks of HIIT, the T lymphocytes subset distribution returned to pre-train values. Heavens et al. [31] selected resistance-trained individuals for a CrossFit^®^ protocol that consisted in 10 repetitions, with a decrease of one repetition after each set until, reaching one repetition at 75% 1RM in powerlifting exercises. The study found changes in IL6, myoglobin, and creatine kinase 60 min post-exercise, which led the authors to conclude that there should be a decrease in intensity or an increase in rest days between sessions.

Comparing men and women, men showed a higher testosterone response to weightlifting exercises than women [32]. Kraemer and Ratamess [26] stated that testosterone increased in men and women due to different neuromuscular, morphologic, and metabolic actions, in which men experience the action of testosterone, and women that of estrogen. The direct action of these hormones influences the cellular composition of muscles, as men have a greater muscle mass than women and despite the similarities in the muscle fibers, men have greater quantities of type II fibers. These characteristics explain why men have greater values of testosterone in comparison with women. Cortisol and immunological responses did not change, probably because the stress caused by the training was the same in both genders. These results are in agreement with higher CD4 values than CD8 values in healthy adults [33]. However, in other type of athletes such as elite swimmers [34] or Taekwondo athletes [35], the CD4/CD8 ratio was inverted.

## 5. Conclusions

This study analyzed the chronic hormonal and immune system responses after six months of CrossFit^®^ training, comparing male and female practitioners. CrossFit^®^ training changed the serum and basal levels of testosterone and cortisol in men (with an increase in testosterone and a decrease in cortisol). In CD4 T lymphocytes, no significant changes were observed after six months of CrossFit^®^ training in both genders. However, there was a decrease in CD8 T lymphocyte basal values during the first four months of CrossFit^®^ training, with a return to the baseline values after six months in both genders, suggesting that there is an adaptation of the immune system to this type of training.

## Figures and Tables

**Table 1 ijerph-16-02577-t001:** Training sessions distribution per exercise modality.

Week	Monday	Tuesday	Wednesday	Thursday	Friday	Saturday	Sunday
Week 1	M	G + W	M + G + W	M + G	W	Rest	Rest
Week 2	G	M + W	M + G + W	G + W	M	Rest	Rest
Week 3	W	M + G	M + G + W	M + W	G	Rest	Rest

M: Metabolic conditioning exercises; G: Gymnastics exercises; W: Weightlifting exercises.

**Table 2 ijerph-16-02577-t002:** Participants’ anthropometric measurements and practice time.

Variables	Men	Women	*p*-Value
Age (years)	34.7 ± 7.5	36.1 ± 13.6	0.7417
Height (cm)	180.7 ± 4.8	166.8 ± 7.5	0.0000 ^†^
Total Body Mass (kg)	89.2 ± 7.1	65.5 ± 15.7	0.0000 ^†^
Estimated Body Fat (%)	17.6 ± 2.1	23.5 ± 4.8	0.0013 ^†^
Practice Time (months)	9.4 ± 2.7	8.8 ± 1.9	0.5076

^†^ Difference between men and women.

**Table 3 ijerph-16-02577-t003:** Mean ± standard deviation of testosterone (T) and cortisol (C) levels and the ratio of testosterone to cortisol (T/C) at different measurement times in men and women.

Variable	T0	T2	T4	T6
**Total**				
Testosterone (pg·mL^−1^)	261.7 ± 249.3	289.5 ± 232.6	298.7 ± 258.5	346.0 ± 299.7 * ^0,2,4, a^
Cortisol (pg·mL^−1^)	18.0 ± 8.2	18.1 ± 9.7	15.6 ± 6.3 * ^0, c^	14.6 ± 5.6 * ^0,2, d^
T/C	18.1 ± 19.7	21 ± 20.3	23 ± 22.8	28.1 ± 27.2
**Men**				
Testosterone (pg·mL^−1^)	421.2 ± 207.6 ^††^	451.4 ± 143.9 ^††^	473.7 ± 189.4	564.7 ± 185.3 * ^0,2,4, ††, b^
Cortisol (pg·mL^-1^)	15.6 ± 5.2 ^†^	15 ± 5.4 ^†^	13.3 ± 4.3 * ^0, †, e^	12.6 ± 3.4 * ^0,2,4, †, e, f †, g^
T/C	30.4 ± 18	34.6 ± 16.2	37.8 ± 19.2	47.6 ± 19.6
**Women**				
Testosterone (pg·mL^−1^)	35.7 ± 21.9	60.2 ± 99	50.8 ± 60	36.3 ± 14.1
Cortisol (pg·mL^−1^)	22.6 ± 9.1	23.8 ± 11.3	19.6 ± 6.5	18.3 ± 60
T/C	1.7 ± 0.9	2.7 ± 3.6	3.2 ± 4.8	2.1 ± 0.9

* 0,2,4 Difference compared to T0, T2, and T4, respectively (*p* < 0.05); ^†^ Difference compared to women (*p* < 0.05); ^††^ Difference compared to women (*p* < 0.0001); ^a^ T0 (*p* = 0.001), T2 (*p* = 0.002), and T4 (*p* = 0.001); ^b^ T0 (*p* = 0.001), T2 (*p* = 0.003), and T4 (*p* = 0.001); ^c^ T0 (*p* = 0.022); ^d^ T0 (*p* < 0.0001) and T2 (*p* = 0.00); ^e^ T0 (*p* = 0.049); ^f †^ T0 (*p* = 0,028), T2 (*p* = 0.013), T4 (*p* = 0.002), and T6 (*p* = 0.002); ^g^ T0 (*p* = 0.003) and T2 (*p* = 0.023).

**Table 4 ijerph-16-02577-t004:** Mean ± standard deviations of lymphocytes CD4 and CD8 values at different measurement times.

Variable	T0	T2	T4	T6
**Total**				
CD4 (cells/mm^3^)	1100.5 ± 307.0	1026.9 ± 305.6	1045.7 ± 275.5	1118.8 ± 242.2
CD8 (cells/mm^3^)	664.9 ± 220.9	582.3 ± 226.4	623.4 ± 195.4	672.4 ± 196.9 * ^2,4^
CD4/CD8	1.66 ± 1.39	1.76 ± 1.35	1.68 ± 1.41	1.66 ± 1.23
**Men**				
CD4 (cells/mm^3^)	1108.1 ± 270.5	1065.8 ± 328.0	1053.7 ± 284.8	1126.2 ± 249.2
CD8 (cells/mm^3^)	723.0 ± 231.9	659.4 ± 243.8 ^†^	673.2 ± 208.2	728.3 ± 208.5 * ^2,4^
CD4/CD8	1.53 ± 1.17	1.62 ± 1.35	1.57 ± 1.37	1.55 ± 1.19
**Women**				
CD4 (cells/mm^3^)	1089.6 ± 365.0	971.7 ± 275.0	1034.4 ± 274.0	1108.3 ± 242.6
CD8 (cells/mm^3^)	582.5 ± 182.9	473.1 ± 147.5	553.0 ± 157.8	593.1 ± 154.0 * ^2^
CD4/CD8	1.87 ± 2.0	2.05 ± 1.86	1.87 ± 1.74	1.87 ± 1.58

* ^2,4^ Difference compared to T2 and T4, respectively (*p* < 0.05); ^†^ Difference compared to women (*p* < 0.05).

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
