# Peer review of "Gender Differences in Chronic Hormonal and Immunological Responses to CrossFit®"

_ijerph, 2019, doi:10.3390/ijerph16142577_

Round 1

Reviewer 1 Report

I have read with great interest the article by authors regarding to analyze the chronical responses of hormonal and immune systems after a CrossFit® training period, to allow prescribe exercise for both genders. For reason this study seems relevant.

I believe there is a very good contribution for the current state of the art in this specific topic. The clarity and flow in some parts still need minor improvement.

Minor comments

Abtract

page 1

In the abstract no reference is made to the results of T CD4  lymphocytes response to exercise.

line 24 -  indicate the units of measurement

page 2

line 1 - scale reference?

2. Materials and Methods

Sample

line 27 - indicate in the inclusion criteria the minimum assiduity rate accepted for the subjects to integrate the sample

line 10 - it is suggested that the results of the study by Jin et al. [13] are related to acute effects of the exercise.

page 3

line 8 - I suggest to remove reference number 17 because the study refers to cancer survivors, and the sample of the present study are healthy subjects

line 11 - the authors say that participants were told "were they asked to refrain from doing any physical exercise outside the study, but were allowed to continue their life as usual freely", and what indications were given regarding daily diet?

- line 14 - we suggest that the authors justify the intensity categories (RPE between 8 and 10 in the OMNI-RES) to be reached during WOD, since the intensities and scale used in the study of Ratamess et al. (2008), the intensities are lower and the scale used is not indicated (Borg CR-10 or OMNI-RES 0-10 scale). As a recommendation for future studies we suggest the quantification of the training load of each session, e.g., determine by the product of s-RPE  for the session' duration.

line 26 - indicate how the T CD4 and CD8 lymphocytes were counted

page 4

 -table 3 - indicate the units of measure of the parameters

page 5

 -table 4 - indicate the units of measure of the parameters (cells/mm3, %, !?)

In order to compliment the discussion the authors could have determined and analyzed the CD4: CD8 ratio, since it is a measure of how the immune function is balanced.

Author Response

Dear Reviewer,

We appreciate the interest that the editors and reviewers have taken in our manuscript and the constructive criticism they have given. We have addressed all the minor concerns of the reviewers. We have included a point-by-point response to the reviewers in addition to making the changes described above in the manuscript. Changes to the text and footnotes of the tables in the manuscript are marked in red.

Minor Comments

Abtract

page 1

In the abstract no reference is made to the results of T CD4  lymphocytes response to exercise.

ANSWER: DONE. We have included in the manuscript that there are no differences in T CD4

line 24 -  indicate the units of measurement

ANSWER: DONE

page 2

line 1 - scale reference?

DONE

2. Materials and Methods

Sample

line 27 - indicate in the inclusion criteria the minimum assiduity rate accepted for the subjects to integrate the sample

ANSWER: DONE. We have included  in the manuscript a minimum attendance of 85% of all sessions

page 3

line 8 - I suggest to remove reference number 17 because the study refers to cancer survivors, and the sample of the present study are healthy subjects

ANSWER: DONE.

line 11 - the authors say that participants were told "were they asked to refrain from doing any physical exercise outside the study, but were allowed to continue their life as usual freely", and what indications were given regarding daily diet?

ANSWER: DONE. We have included  in the manuscript they were allowed to continue their daily diets.

- line 14 - we suggest that the authors justify the intensity categories (RPE between 8 and 10 in the OMNI-RES) to be reached during WOD, since the intensities and scale used in the study of Ratamess et al. (2008), the intensities are lower and the scale used is not indicated (Borg CR-10 or OMNI-RES 0-10 scale). As a recommendation for future studies we suggest the quantification of the training load of each session, e.g., determine by the product of s-RPE  for the session' duration.

ANSWER: DONE. We have included  in the manuscript the reference. Thank your for the recommendation on quantification of the training load, we will do it in future works.

line 26 - indicate how the T CD4 and CD8 lymphocytes were counted

ANSWER: DONE. 

page 4

 -table 3 - indicate the units of measure of the parameters

ANSWER: DONE. 

page 5

 -table 4 - indicate the units of measure of the parameters (cells/mm3, %, !?)

ANSWER: DONE. 

In order to compliment the discussion the authors could have determined and analyzed the CD4: CD8 ratio, since it is a measure of how the immune function is balanced.

ANSWER: DONE. We have included  in the manuscript the ratio and some studies in the discussion.

Thank you again for consideration of our revised manuscript. 

Best regards, 

Reviewer 2 Report

Explain in the introduction why you think it would be differences by sex, there is in the title but not in the objective of the study. Idem in the discussion. 

tables 3 and 4 - include the unit of the variables

Discuss your data with previous studies in strength and endurance based sports. why CrossFit would elicit a different physiological response??

Author Response

Dear Reviewer,

We appreciate the interest that the editors and reviewers have taken in our manuscript and the constructive criticism they have given. We have addressed all the minor concerns of the reviewers. We have included a point-by-point response to the reviewers in addition to making the changes described above in the manuscript. Changes to the text and footnotes of the tables in the manuscript are marked in red.

Minor Comments

POINT 1. tables 3 and 4 - include the unit of the variables

RESPONSE 1: DONE

POINT 2. Explain in the introduction why you think it would be differences by sex, there is in the title but not in the objective of the study. Idem in the discussion. 

RESPONSE 2: DONE. We have included  in the introduction: "Between men and women, few studies have analyzed these responses with CrossFit® training. Murawska-Cialowicz et al. [14], verified at rest changes in brain-derived neurotrophic factor and in irisin levels and improvements in aerobic capacity and body composition of young physically active men and women. However, in testosterone (T), cortisol (C) and T/C ratio were observed similar results between men and women, and between gender and time of exercise. and "We hypothesized that (a) there will be changes in the chronical responses of hormonal and immune systems after a six-months CrossFit® training intervention, (b) only men will present changes in the chronical hormonal responses, and c) no differences will be observed between genders in chronical immunological responses."

POINT 3. Discuss your data with previous studies in strength and endurance based sports. why CrossFit would elicit a different physiological response??

RESPONSE 3: DONE. We have included  in the discussion: "Comparing men and women, men have a higher testosterone response in weightlifting exercises than women [33]. Kraemer and Ratamess [27] stated that testosterone increased in men and women due to different neuromuscular, morphologic and metabolic actions, in which men experience the action of testosterone and women estrogen. The direct action of these hormones influences the cellular composition, as men have a greater muscle mass than women, and despite the similarities in the muscle fibers men have greater quantities of type II fibers. These characteristics explain why men have greater values of testosterone in comparison with women. Cortisol and immunological responses did not change, probably, due to the stress of the training being the same between genders. Results that are in agreement with higher CD4 values compared toCD8 in healthy adults [34]. However, in other type of athletes such as elite swimmers [35] or Taekwondo athletes [36], the CD4:CD8 ratio was inverted."

Thank you again for consideration of our revised manuscript. 

Best regards, 

Reviewer 3 Report

Dear authors,

I want to congratulate to authors because you have realized an interesting research making the first research that analyse the chronic effect of hormonal and immunological responses to a Crossfit training program. Also the methods are very good and the presentation of the information in the paper is excellent. Nevertheless, I think that authours would realise the next suggestions:

-          At the final of the first paragraph would be interesting include a mention to the excessive training load like a possible mechanism of the frequent prevalence of sport injuries in crossfit athletes.

-          Does some performance value like inclusion criteria?

-          In the expressed results of testosterone, cortisol and ratio TC would be interesting to include the differences between the different moments (I suggest to include like %).Also in the Table 3 it would be interesting to include the p-value of the ANOVA.

-          In the Table 4 I think that it´s necessary to include the p-value of moment, gender and interaction.

-          Conclusions: it´s necessary to remove “We have conclude”. It´s necessary to avoid write in first person.

Author Response

Dear Reviewer,

We appreciate the interest that the editors and reviewers have taken in our manuscript and the constructive criticism they have given. We have addressed all the minor concerns of the reviewers. We have included a point-by-point response to the reviewers in addition to making the changes described above in the manuscript. Changes to the text and footnotes of the tables in the manuscript are marked in red.

Minor Comments

POINT 1. At the final of the first paragraph would be interesting include a mention to the excessive training load like a possible mechanism of the frequent prevalence of sport injuries in crossfit athletes.

RESPONSE 1. DONE. We have included  in the introduction: "The training loads used are not individualized which can be excessive for some individuals and can lead to increased risk of injury [2]." 

POINT 2.   Does some performance value like inclusion criteria?

RESPONSE 2. DONE. No, all of the subjetcs of our sample had a similiar performance and had a minimum experience of 6 months

POINT 3.  In the expressed results of testosterone, cortisol and ratio TC would be interesting to include the differences between the different moments (I suggest to include like %).Also in the Table 3 it would be interesting to include the p-value of the ANOVA.

RESPONSE 3. DONE. We have included the p-value in table 3 and % in the results.

POINT 4. In the Table 4 I think that it´s necessary to include the p-value of moment, gender and interaction.

RESPONSE 4. DONE. We have included the p-value in table 4

POINT 5.  Conclusions: it´s necessary to remove “We have conclude”. It´s necessary to avoid write in first person.

RESPONSE 5. DONE. 

Thank you again for consideration of our revised manuscript. 

Best regards, 

Round 2

Reviewer 2 Report

Congratulation for the nice work conducted